# The Venom Composition of the Snake Tribe Philodryadini: ‘Omic’ Techniques Reveal Intergeneric Variability among South American Racers

**DOI:** 10.3390/toxins15070415

**Published:** 2023-06-27

**Authors:** Emilly Campos Tioyama, Juan David Bayona-Serrano, José A. Portes-Junior, Pedro Gabriel Nachtigall, Vinicius Carius de Souza, Emidio Beraldo-Neto, Felipe Gobbi Grazziotin, Inácio L. M. Junqueira-de-Azevedo, Ana Maria Moura-da-Silva, Luciana Aparecida Freitas-de-Sousa

**Affiliations:** 1Programa de Pós-Graduação em Ciências-Toxinologia, Escola Superior do Instituto Butantan, São Paulo 05508-210, Brazil; emilly.tioyama.esib@esib.butantan.gov.br (E.C.T.); juan.serrano.esib@esib.butantan.gov.br (J.D.B.-S.); 2Laboratório de Imunopatologia, Instituto Butantan, São Paulo 05503-900, Brazil; portes.junior@butantan.gov.br (J.A.P.-J.); ana.moura@butantan.gov.br (A.M.M.-d.-S.); 3Laboratório de Toxinologia Aplicada, Instituto Butantan, São Paulo 05503-900, Brazil; pedronachtigall@gmail.com (P.G.N.); vinicius.souza.esib@esib.butantan.gov.br (V.C.d.S.); inacio.azevedo@butantan.gov.br (I.L.M.J.-d.-A.); 4Laboratório de Bioquímica e Biofísica, Instituto Butantan, São Paulo 05503-900, Brazil; emidio.beraldo@butantan.gov.br; 5Laboratório de Coleções Zoológicas, Instituto Butantan, São Paulo 05503-900, Brazil; felipe.grazziotin@butantan.gov.br

**Keywords:** venom composition, *Philodryas*, snake endogenous matrix metalloproteinases type 9 (seMMP-9), enzymatic assays

## Abstract

Snakes of the Philodryadini tribe are included in the Dipsadidae family, which is a diverse group of rear-fanged snakes widespread in different ecological conditions, including habitats and diet. However, little is known about the composition and effects of their venoms despite their relevance for understanding the evolution of these snakes or even their impact on the occasional cases of human envenoming. In this study, we integrated venom gland transcriptomics, venom proteomics and functional assays to characterize the venoms from eight species of the Philodryadini tribe, which includes the genus *Philodryas*, *Chlorosoma* and *Xenoxybelis*. The most abundant components identified in the venoms were snake venom metalloproteinases (SVMPs), cysteine-rich secretory proteins (CRISPs), C-type lectins (CTLs), snake endogenous matrix metalloproteinases type 9 (seMMP-9) and snake venom serinoproteinases (SVSPs). These protein families showed a variable expression profile in each genus. SVMPs were the most abundant components in *Philodryas*, while seMMP-9 and CRISPs were the most expressed in *Chlorosoma* and *Xenoxybelis*, respectively. Lineage-specific differences in venom composition were also observed among *Philodryas* species, whereas *P. olfersii* presented the highest amount of SVSPs and *P. agassizii* was the only species to express significant amounts of 3FTx. The variability observed in venom composition was confirmed by the venom functional assays. *Philodryas* species presented the highest SVMP activity, whereas *Chlorosoma* species showed higher levels of gelatin activity, which may correlate to the seMMP-9 enzymes. The variability observed in the composition of these venoms may be related to the tribe phylogeny and influenced by their diets. In the presented study, we expanded the set of venomics studies of the Philodryadini tribe, which paves new roads for further studies on the evolution and ecology of Dipsadidae snakes.

## 1. Introduction

The snake family Dipsadidae belongs to a very diverse clade known as Caenophidia (advanced snakes), and with approximately 700 described species [1], this family accounts alone for 20% of the extant diversity of snake in the world [2]. Dipsadids usually present opisthoglyphous maxillary dentition, which is characterized by grooved teeth located in the posterior region of the mouth, in association with venom glands [3,4]. Despite producing a complex set of toxins [5], Dipsadidae has not been comprehensively studied as other snake families have, such as Viperidae and Elapidae [6]. Human envenomation by dipsadids is seldom life threatening, although it is characterized by local pain, edema, local bleeding, slight increase in temperature at the site of the bite, erythema and ecchymosis [7].

Within dipsadids, species of the South American tribe Philodryadini Cope, 1886 [2] present a wide geographical distribution, occurring from Patagonia to the northern Amazon Forest [8,9,10]. Philodryadini includes the following four genera: *Philodryas* Wagler, 1830 (16 species); *Ditaxodon* Hoge, 1958 (01 species); and the newly revalidated genera *Xenoxybelis* Machado, 1993 (02 species) and *Chlorosoma* Wagler, 1830 (02 species) [4]. Most of these snakes can be divided in two distinct assemblages: (1) *Philodryas* and *Ditaxodon*, in which most of the species are active during the day, foraging in open areas and mainly presenting terrestrial habits (although some species such as *P. olfersii* are mostly arboreal); and (2) *Xenoxybelis* and *Chlorosoma*, which are diurnal forest dwellers, being considered exclusively arboreal species [8,9,10].

Regarding their feeding ecology, several studies have indicated that most species of Philodryadini have generalist diets, feeding on invertebrates, small rodents, lizards, amphibians, birds [9,10,11,12,13,14,15,16,17,18,19,20], bats [21,22] and other snakes [19,23]. On the other hand, some species, such as *P. agassizii*, differ by having a very specialized diet. Particularly, *P. agassizii* is singular among dipsadids because it almost exclusively preys on spiders of the genus *Lycosa* and occasionally ingests scorpions and lizards [24]. Some studies have proposed that in snakes, the venom and diet had evolved together [25,26], suggesting that diverse diets—such as shown by Philodryadini—are associated with diverse venoms [27].

Snake venoms are mixtures formed by peptides, proteins, carbohydrates and other bioactive molecules [28]. They are important for the immobilization and possible digestion of prey and are also used against potentially threatening predators [29,30,31]. Interest in snake venom composition has increased in recent years, being of paramount importance to understand the mechanisms of toxin evolution and the comprehension of clinical effects and to improve the development of antidotes [26]. “Omics” technologies have considerably advanced in recent years, enabling qualitative and quantitative progress in the study and characterization of biochemical compounds [5]. These improvements provided by “Omics” technologies have also contributed to the understanding of the levels of biological variability and to the inference of the processes that generate global biodiversity [5,25,26,32].

Several studies have shown that venom variability can be observed at different biological levels, such as interspecific [33], intraspecific [34,35], geographical [36,37,38,39,40], sexual [41], ecological [25,42,43,44], allometric [35] and ontogenetic levels [43,45,46,47]. The differences among these venoms are usually explained by factors that modulate the expression of toxin-related genes, although there are other mechanisms such as gene duplication, gene recruitment, alternative splicing and neofunctionalization of genes [25,48,49,50,51] involved in the generation of the extensive diversity of snake venoms. The association between variability of function and composition among different biological levels makes venoms an excellent model to explore ecological and evolutionary processes in snakes.

Considering Philodryadini, few studies on composition, isolation and characterization of toxins have been performed. *Philodryas olfersii* represents the best-studied species in this tribe [52,53,54]. From the venom of *P. olfersii*, one myotoxin [52] and five fibrinogenolytic proteases were already isolated [53]. Of these five fibrinogenolytic proteases, four are snake venom metalloproteinases (SVMPs) and one is a snake venom serine proteinase (SVSP) [53]. Transcriptome analyses of the venom gland of *P. olfersii* confirmed the presence of SVMPs and SVSPs and described the presence of cysteine-rich secretory proteins (CRISPs), C-type lectins (CTLs) and C-type natriuretic peptide (CNP) [54].

Besides *P. olfersii*, the only other species of Philodryadini that is considerably well studied regarding venom composition is *P. patagoniensis*. Two toxins have already been isolated from the venom of *P. patagoniensis*, one SVMP and one CRISP. The SVMP was denominated Patagonfibrase, and presents hemorrhagic, inflammatory, fibrinogenolytic and azocaseinolytic activity [55,56,57]. The CRISP was denominated Patagonin [58] and shows myotoxic and antimicrobial activities [59]. Considering the ecological and biological diversity of a group such as Philodryadini—composed of 4 genera and 21 species—it is noteworthy that the available knowledge of venom composition for this tribe is based mostly only on two species, lacking information about all other Philodryadini.

In this context, we substantially expand the knowledge about the composition of venoms of Philodryadini. We provide the biochemical (transcriptomes and proteomes) and functional (biological essays) characterizations of the venom of eight species of Philodryadini, representing three genera (*Chlorosoma*, *Philodryas* and *Xenoxybelis*). The venom composition and function of six species is presented here for the first time. Additionally, we compare the venom variability of these snakes with their phylogeny, habitat and diet, expanding the understanding of venom evolution of “rear-fanged” snakes.

## 2. Results

### 2.1. Philodryadini Duvernoy’s Venom Gland Transcriptome and Venom Composition

For transcriptomic analysis, we sequenced the venom gland mRNA transcriptome of at least one representative individual for each species with Illumina technology. The number of reads ranged from approximately 2.4 million for *P. olfersii* (SB0001) to almost 24 million reads for *P. patagoniensis* (Appendix A). A final master set with the toxin-coding sequences from all species was curated and resulted in 251 different toxin transcripts from several toxin families (Appendix A). In general, the families of proteins related to snake toxins expressed in the venom glands in our study were 3FTX (three-finger toxin), 5NUCL (5-nucleotidase), AChE (acetylcholinesterase), CNP (C-type natriuretic peptide), CRISP (cysteine-rich secretory protein), CTL (C-type lectins), CVF (cobra venom factor), CYS (cystatin), FAXV (snake venom coagulation factor X like—belonging to the S1 family of serine peptidase), FICOLIN (ficolin), HYAL (hyaluronidase), KUNZ (BPTI/KUNITZ inhibitor domain), KUWAP (Ku-wap-fusin), iPLA (phospholipase inhibitor), PDE (phosphodiesterase), PLB (phospholipase B), seMMP-9 (snake endogenous matrix metalloproteinase 9), SVLIPA (snake venom acid lipase), PIII-class SVMPs (snake venom metalloproteinase PIII-class), SVSP (snake venom serine proteinase), VEGF (vascular endothelial growth factor) and WAP (waprin whey acidic protein-type) (Appendix A). However, some of these protein families were found at very low levels of expression, indicating that they may not represent actual toxins in these snakes.

On average, approximately 18% of all transcripts corresponded to toxins in the transcriptomes; the largest percentage of transcripts predicted for toxins was found in *P. olfersii* (SB0001) with 34%, and the individual with the lowest percentage of predicted transcripts was *X. argenteus* with 8.5% (Figure 1; Appendix A). In general, the most expressed toxins in the venom glands were PIII-class SVMPs, CRISPs, seMMP-9 and CTLs. Surprisingly, according to the abundance and distribution of each component expressed in the venom gland, we observed a very different expression profile for each genus (Figure 1; Appendix A). SVMPs were the most abundant components in *Philodryas*, while seMMP-9 were the most expressed in *Chlorosoma* and CRISPs in *Xenoxybelis.*

The three individuals of *P. olfersii* had the highest proportions of transcripts for PIII-class SVMPs, with more than 63% of the total toxins represented by these toxins. *P. mattogrossensis*, *P. patagoniensis*, *P. nattereri* and *P. agassizii* presented 58.4%, 52.6%, 45.9% and 39.1% of transcripts for PIII-class SVMPs, respectively. Nonetheless, *C. viridissimum*, *C. laticeps* and *X. argenteus* showed much lower levels of PIII-class SVMPs, with 27.3%, 13.9% and 25.5%, respectively. Interestingly, predicted transcripts for seMMP-9 predominated in *Chlorosoma* species, corresponding to 56.2% of the total transcripts for *C. laticeps* and 41.6% for *C. viridissimum*. On the other hand, transcripts corresponding to CRISPs were the main component being found in the venom gland of *X. argenteus,* with 56.9% of expression. In the other species, CRISPs were expressed in 29.8% of the individual of *P. nattereri*, 20.6% in *P. olfersii* (SB0132), 20.2% in *P. agassizii*, 19.3% in *P. patagoniensis*, 16.6% in *P. olfersii* (SB0026), 13.2% in *C. viridissimum*, 12.10% in *P. olfersii* (SB0001) and lowly expressed in the individual of *C. laticeps* with 3.5%. CTL transcripts were also expressed in all individuals, with the highest expression in the specimen of *C. laticeps* at 16.1%, followed by *P. agassizii* at 15.7%, *P. nattereri* at 12.4% and *P. olfersii* (SB0001) with 11.4%. SVSPs were not detected in the glands of *P. agassizii*, *P. mattogrossensis* and *X. argenteus* but were more expressed in the two individuals of *Chlorosoma* and in the individual of *P. patagoniensis*, delineating an expression pattern with similar amounts in the three individuals of *P. olfersii*. The last component that showed reasonable levels of expression (0.1% to 6%) was the CNPs. They were more expressed in all *P. olfersii* individuals and were not detected only in *P. agassizii* and *X. argenteus*. All individuals presented other toxin components lowly expressed, such as 5NUCL, AChE, CYS, CVF, FICOLIN, HYAL, KUNZ, KUWAP, iPLA, PDE, PLB, SVLIPA, VEGF and WAP.

According to the abundance and distribution of each component expressed in the venom gland, the families of toxins showed an expression profile similar within each genus analyzed. We also noticed a low intraspecific variability in *P. olfersii* individuals although these individuals were from distinct sexes and geographical origin. However, quantitative variations were observed in the transcriptomic profiles among the *Philodryas* genera. *P. mattogrossensis* presented a more simplified profile compared to the other species, which was predominantly composed of PIII-class SVMPs, CRISPs and CTLs.

Interestingly, *P. agassizii* presented the most distinct profile, with 3FTXs being one of the highly expressed transcripts (around 20.2% of toxin transcripts). They are commonly found in elapid venoms but can be detected in colubrids [60]. 3FTXs were also detected in less than 2% of the venom glands of *X. argenteus*, *C. laticeps* and *C. viridisismum*.

Proteomic analysis of venom pools of Philodryadini was used to validate the existence and abundance of predicted proteins from the transcriptomes of the species of *Philodryas* (*P. olfersii*, *P. patagoniensis*, *P. nattereri*, *P. agassizii*), *Chlorosoma* (*C. laticeps* and *C. viridissimum*) and *Xenoxybelis* (*X. argenteus*). Unfortunately, we were not able to perform proteomic analysis for the *P. mattogrossensis* species because we had an insufficient amount of venom for the analyzed individual. The database used for proteomic analysis was manually compiled from the translation of the annotated putative sequences of the transcriptome of all individuals, forming a single bank for the Philodryadini tribe. The proteins identified were seMMP-9, PIII-class SVMPs, SVSPs, CRISPs, CTLs, 3FTXs, CNP, CVF, FICOLIN, FAXV, KUNZ, KUWAP, iPLA and PLB (Figure 2; Appendix A).

As shown in Figure 2, the data from proteomic analysis of the venoms of *Philodryas* and *X. argenteus* show that PIII-class SVMPs gain prominence, equivalent to more than half of the total of toxins found, which confirms the high expression of these transcripts in the venom glands. CRISPs were the second-largest component of the venom found in in most species. The largest proportions were found in *P. olfersii* (25.5%) and *X. argenteus* (22.5%), followed by *P. agassizii* (10.2%) and *P. nattereri* (9.5%).

A curious result that attracted our attention was that seMMP-9 was evidenced in venoms in comparison with the transcriptome profile of the glands. There was an increase in *P. patagoniensis* (27.4%) and *P. nattereri* (19.7%) and even in individuals who presented minimal expression of this component in the transcriptome, such as *P. olfersii*, *P. agassizii* and *X. argenteus*. These toxins are more abundant in the venoms of *Chlorosoma,* showing to be a characteristic of the genus that differs from the others. CTLs were also present in all venoms, however, in smaller amounts, at a ratio of 0.8–2.7%. 3FTXs were expressed only in *P. agassizii*, confirming the existence of these toxins in the venom, as well as CNPs. The SVSPs were found at more significant levels only in *P. olfersii* (7.1%) and *C. laticeps* (5.5%) venom proteomes.

The discrepancies observed between transcription and venom abundance, such as PIII-class SVMPs in the venoms of *P. agassizii* (39.1%:76.7%) and seMMP-9 in *C. viridissimum* (41.6%:74%), have already been found in other species, such as *Bothrops atrox* [34]. These authors suggest that changes in mRNA/protein turnover may act differentially on each transcript within a single toxin family. However, it is necessary to analyze a larger number of individuals for each species to confirm this hypothesis.

To confirm the profile obtained using the transcriptome of venom glands and the proteome of venom, we also performed an electrophoretic profile of Philodryadini venoms (Figure 3). The electrophoretic profile of Philodryadini venoms shows proteins of molecular mass in the range of 14.4 kDa to 97 kDa (Figure 3A). A common feature among them is that all venoms showed bands in the 50–80 kDa region, which is known to represent the molecular mass of snake venom metalloproteinases (SVMPs), an important component found in venom of most venomous snakes [61,62]. In this region, venoms from *P. nattereri* and *Bothrops jararaca* (control) show a high hydrolytic activity (Figure 3B). By shotgun mass spectrometry identification of crude venoms, SVMPs were predominant in this range (Appendix A). Bands in the region of 27–30 kDa can also be seen in all venoms, and this range of molecular mass corresponds to cysteine-rich secretory proteins (CRISPs) [58]. In the 14.4 kDa region, double bands more evident in *Chlorosoma* may correspond to the α and β subunits of C-type lectins (CTLs), with molecular weight of 14–15 and 13–14 kDa, respectively [63]. Regarding the observed differences, in the region of 35–40 kDa, stronger bands are seen in *C. viridissimum*, being the two venoms of the genus *Chlorosoma* not completely inhibited by EDTA. In addition to the proteomic and transcriptomic profile, this result also suggests that the venom composition of the two *Chlorosoma* species differs from *Philodryas* venoms.

### 2.2. Functional Analysis of Philodryadini Venoms

The venom profile observed led us to perform functional assays to test the putative effects of the venom of the Philodryadini tribe. Based on the data of the venom composition of *Philodryas* and *Chlorosoma*, SVMPs, seMMPs-9 and SVSPs were the most abundant enzymatic toxins in the venoms of the analyzed species. Thus, we evaluated the enzymatic activity of these venom pools using specific substrates for metalloproteinases (SVMPs and seMMPs-9) and serinoproteinases (SVSPs). The venoms of *P. agassizii*, *P. mattogrossensis* and *X. argenteus* were not included in the assay due to insufficient amounts. The results showed that venoms from *Philodryas* showed activity in the substrate for metalloproteinases, however, the activity was considered low when compared to the venom of *Bothrops jararaca*, used as a positive control (Figure 4A). As expected, the venom of *P. olfersii* was the only one showing SVSP activity (Figure 4B), however, the SVSP activity was lower than that observed in the activity of *Bothrops neuwiedi* venom, used as a positive control. The enzymatic activity observed here corroborates the results of Peichoto et al. (2012) [64] showing that the venom of *P. olfersii* exhibited the highest level of catalytic activity for the serine proteinase substrate among four species of opisthoglyph snakes, including *P. baroni* and *P. patagoniensis*. We also observed that in the plate gelatinolytic assay (Figure 4C), the Philodryadini venoms showed a high proteolytic capacity, even in a smaller concentration used for the comparison with the *B. jararaca* venom, the positive control. This result also suggests that the best substrate for metalloproteinases in Philodryadini venoms would be gelatin and not the synthetic Abz-ALGLA-EDDnp peptide substrate.

To confirm the gelatinolytic activity of metalloproteinases (SVMPs and seMMPs) present in Philodryadini venoms, we performed a zymography assay on a 12.5% SDS-PAGE prepared in the presence of gelatin with and without the protease inhibitor EDTA (Figure 3C). We used *B. jararaca* venom as a positive control, well established in the literature for being rich in SVMPs [65,66]. According to the results obtained, all tested venoms were able to degrade gelatin, as shown by white or transparent bands on the blue background of the gel, showing that *Philodryas* and *Chlorosoma* venoms have a potent gelatinolytic character, an expected result due to the high abundance of PIII-class SVMPs and considerable amounts of seMMPs-9 present in the transcriptome and proteome of these snakes. In Figure 3A, we show the electrophoretic resolution of the major toxins of the five venoms, with a marked presence of bands above 50 kDa in all venoms and the predominance of bands between 35 and 40 kDa in *Chlorosoma viridissimum* venom. In Figure 4B, clear bands can be seen in all venoms in the 35–40 kDa range. In addition, in *P. nattereri* venom, a large white band can be seen between 36–66 kDa, which is in agreement with the enzymatic assay for SVMPs, where it showed greater activity compared to other Philodryadini snakes. The other venoms, on the other hand, showed strong white bands only in the 35 kDa region. The only compatible proteinases at this molecular mass range in those venoms are seMMPs since P-III SVMPs are larger and P-I and P-II SVMPs are not present in Dipsadidae snake venoms [5,49,67] as well as not found in our transcriptome and proteome results. In addition, the incubation of venoms with 20 mM of the metal chelator EDTA, a metalloproteinase inhibitor (Figure 3C), inhibited almost completely the hydrolysis of gelatin, as almost all white bands disappeared, except for the 35 kDa bands from *Chlorosoma* (indicated by the black arrow on Figure 3C).

To test the functional roles of seMMPs in the gelatinolytic activity, the 35–40 kDa bands of all venoms were excised from the gel and submitted to mass spectrometry (Appendix A) to identify the enzymes that are exerting strong proteolysis even in the presence of the EDTA inhibitor. The polypeptides identified with the highest score in each band are shown in Table 1, whereas raw data obtained can be seen in Appendix A.

Most of the identified peptides corresponded to seMMPs in the excised bands (35–40 kDa) from *C. laticeps*, *C. viridissimum*, *P. patagoniensis* and *P. nattereri* venoms, except for *P. olfersii*, which presented SVSP peptides (band 7). Despite the identified SVSP peptides (Appendix A), the high number of seMMP peptides suggests that the high gelatinolytic activity observed in *Chlorosoma* venoms is mainly due to the presence of seMMPs.

To better understand the gelatinolytic capacity of Philodryadini venoms and to confirm which classes of enzymes were responsible for the activity, we performed an assay with fluorescein-conjugated gelatin, this time using the inhibitors 1,10-phenanthroline (metalloproteinase inhibitor) and PMSF (serine proteinase inhibitor) at a concentration of 100 mM. The results again indicate that the high gelatinolytic capacity is due to the action of metalloproteinases since 1,10-phenanthroline was able to almost completely inhibit the activity of the venoms, meanwhile the reduction in gelatinolytic activity after incubation with PMSF, if any, was not statistically significant (Figure 5).

## 3. Discussion

The venom gland transcriptomics combined with venom proteomics have allowed the uncovering of venom composition of several poorly studied rear-fanged snakes [5]. Moreover, the possibility of generating comprehensive datasets containing key information about complex phenotypes—such as venom composition—has opened new ways to evaluate and understand the evolution and ecological adaptability of snakes [68,69,70].

In this study, we provided the most extensive overview about the composition and biological functions of venoms of Philodryadini, a very diverse tribe of South American snakes. We described the complete transcriptome of venom glands from ten individuals (eight species) and the venom proteome of seven species to support our findings.

Components present in the venoms of rear-fanged snakes are generally homologous to those found in Viperidae and Elapidae [71]. One of these homologous components is the SVMPs, which were identified in our results as one of the most expressed toxins in both the proteome and the transcriptome of Philodryadini. SVMPs are also the most abundant venom components for most species of vipers [29]. They are functionally versatile toxins, targeting essential points in the hemostasis control and physiological essential tissues of envenomed organisms [72,73].

Hemorrhage is the most evident effect of SVMPs, and it represents the result of a combined disruption of capillary vessel integrity and impairment of the blood coagulation system [74,75,76]. This combined action of SVMPs results in several main disruptive effects, as follows: (1) the consumption of coagulation plasma factors [72] by the activation of coagulation Factor X [77] or Factor II [78]; (2) fibrino(gen)olytic activity [79]; (3) the damage and binding of the capillary vessels [76,80,81]; (4) the inhibition of platelet aggregation [82]. Within Philodryadini, the presence of SVMPs was identified in all evaluated venoms in our study, being the most abundant component in the venoms of species of *Philodryas* and *Xenoxybelis*. Although commonly abundant, SVMPs classes P-I and P-II were not detected in Philodryadini, and only PIII-class was identified in both transcriptomes and proteomes.

These findings about the presence of SVMPs in Philodryadini corroborate studies suggesting that PIII-class is one of the main venom components of snakes from the family Dipsadidae [5]. The contrast regarding the presence/absence of different classes of SVMPs in vipers and dipsadids may be explained by the recent evolution of PIII-class into classes P-I and P-II. These evolutionary modifications in SVMPs are supposed to have occurred after the split of Viperidae from the remaining Caenophidia [49]. Therefore, the multiple episodes of domain alteration responsible for the generation of classes P-I and P-II had occurred only within vipers [49].

Matrix metalloproteinases (MMPs) were also present as one of the main components of Philodryadini venoms. These metalloproteinases are generally found in vertebrates, and they are responsible for important pathophysiological functions, such as extracellular matrix remodeling after injuries, wound healing, growth regulation, reduction of tumors and bone resorption, among others [83,84,85,86,87]. They can be classified according to the organization of their domains or based on substrate specificity. The following three groups are the most commonly studied MMPs: (1) collagenases (MMP-1, MMP-8, MMP-13), consisting of propeptide domain, catalytic domain and hemopexin; (2) gelatinases (MMP-2, MMP-9), which present three repeats of the fibronectin type 2 domain inserted in the catalytic domain; and (3) matrilysins (MMP-7, MMP-26), which are characterized by the absence of the hemopexin domain [87].

The presence of metalloproteinases that are very similar to MMP-9s in the venoms of rear-fanged snakes has been documented with some frequency in recent years [5,88,89,90]. In some groups of dipsadids, a derived type of MMP-9, named svMMP (snake venom MMP), has been consistently found as a major venom constituent [90]. It is characterized by the absence of the hemopexin domain, which is present in all other vertebrate MMP-9s. However, the endogenous type of MMP-9, with its hemopexin domain, was also found to be present with a low to medium expression level in some venoms of dipsadids, as well as in other tissues, and it was named as seMMP [90]. Analyses of sequences of MMPs found in our study revealed the presence of the hemopexin domain, indicating that these matrix metalloproteinases are more similar to the ancestral form of MMP-9, thus corresponding to seMMP-9. This is the first report of bona fide MMP-9 highly expressed in snakes venom glands, which dominates the venom proteomes of all species of *Chlorosoma*.

Philodryadini is phylogenetically related to three other tribes (Conophiini, Tachymenini and Xenodontini) in which the presence of svMMPs was well characterized [90]. Bayona-Serrano et al. (2020) [90] proposed three independent recruitments of svMMPs from an seMMP-9 ancestor and recognized that an early elevation of seMMP levels in venoms of Dipsadidae likely have preceded the appearance of the derived forms (svMMP-A and svMMP-B). Our finding of high levels of seMMP-9 in Philodryadini suggests a rather complex scenario for Dipsadidae in which three tribes (Conophiini, Tachymenini and Xenodontini) have high amounts of svMMPs, one tribe (Philodryadini) presents high levels of seMMPs, whereas all other tribes show non-significant values of MMPs in their venoms.

Our results support the hypothesis suggesting an ancestral participation of MMP-9 in Dipsadidae venoms. Considering the proposal of Bayona-Serrano et al. (2020) [90], which explains the role of MMPs in the venom of dipsadids, these toxins can be performing a similar function to that of SVMP, regarding their proteolytic character. The high expression of these components seen in the transcriptome of *C. laticeps* and *C. viridissimum* explains their high proteolytic capacity even in the presence of the EDTA inhibitor. The identification of peptides by mass spectrometry in the region of gelatin hydrolysis indicates that seMMPs-9 are the main protagonists for the proteolytic activity, a result reinforced by the gelatinolytic assay using specific inhibitors. Based on that, we also noticed that, even for the species of Philodryadini in which seMMPs-9 have a considerably low proportion, seMMPs-9 are providing to these venoms high proteolytic capacity on gelatin. We suppose that, despite the low venom volume produced by species of Philodryadini—when compared to vipers—they compensated by recruiting MMPs by co-option [91,92]. Thus, their venoms can perform highly proteolytic functions, even in minute volumes, which is a very advantageous strategy regarding prey capture and digestion. Moreover, we speculate that dietary selective pressures in these different tribes of Dipsadidae may be the evolutive driver responsible for the emergence of svMMPs in their toxic arsenal.

CRISPs were the third-most abundant component in the venoms of Philodryadini, being the major component expressed in the venom gland of *X. argenteus*. Although its function and biological role are still not very well understood, studies have shown that CRISPs are associated with inflammation by activating the complement system [93] and calcium channels [94]. Peichoto et al. (2009) have previously identified the presence of CRISPs in *Philodryas* [58]. Necrotic activity on the gastrocnemius muscle was demonstrated in this study, indicating that CRISPs may also present myotoxic activity. Posteriorly, they provided a comparison of venom composition among three species from Argentina, and, based on mass spectrometry and the one-dimensional electrophoretic profile in polyacrylamide (SDS-PAGE), bands with approximately 25–26 kDa were identified as CRISPs [64]. One of these CRISPs, called patagonin, was further purified by Badari et al. (2020) from the venom of *P. patagoniensis*, which also showed myotoxic and antimicrobial activity [59]. They also suggested that patagonin binds to ion channels without presenting platelet aggregation or proteolytic and fibrinogenolytic effects [58]. The complete understanding of the role of CRISPs in the envenomation by species of Philodryadini is still an open question, and further biological essays are nevertheless necessary.

The presence of CTLs was also evidenced in all venoms of Philodryadini, particularly in the transcriptome. These proteins have already been reported as present in the transcripts of the venom glands of several rear-fanged snakes, appearing to be a ubiquitous component in these venoms [5]. They are dimeric proteins that contain a carbohydrate recognition domain, inducing hemagglutination by binding glycoconjugates in erythrocytes [63,95]. A recent study showed that these non-enzymatic components in viper venoms are involved in rapid prey immobilization, inducing platelet aggregation, in addition to acute cerebral ischemia, stimulating prey locomotor activity [96]. However, the functional role of CTLs in venoms of dipsadids is still uncharted.

Our results indicate that SVSPs and 3FTXs were more abundant only in the venoms of some particular species of Philodryadini, such as *P. olfersii* and *P. agassizii*, respectively. SVSPs correspond to almost 7% of the total venom proteome of *P. olfersii*. These toxins are known to promote blood clotting by activating clotting factors, as well as inducing platelet aggregation and direct hydrolysis of fibrinogen into fibrin [96]. Assakura et al. (1994) [53] was the first to show that serine proteinase isolated from the venom of *P. olfersii* presents fibrinogenolytic activity, cleaving Aα and Bβ chains of fibrinogen. We support these findings based on our proteomic analysis of a gel band resulting from the electrophoresis of the venom of *P. olfersii* (band seven, Table 1 and Appendix A). Most of the identified peptides for this band corresponded to SVSPs. We suppose that for *P. olfersii*, the gelatinolytic activity may be performed by both seMMPs and SVSPs (Figure 4B,C).

On the other hand, the presence of 3FTXs was abundant only in the venom gland transcriptome of *P. agassizii*, corresponding to 20% of its expressed transcripts. These toxins belong to a superfamily of non-enzymatic proteins commonly found in Elapid venoms but rarely detected in Dipsadidae [44,60,97]. Highly conserved in their structure but highly variable in sequence, 3FTXs are mainly known for their neurotoxic, cardiotoxic and cytotoxic effects [60,96,97,98,99]. In recent years, some studies have shown that these toxins may be taxon specific, targeting specific types of prey [98]. Therefore, not surprisingly, *Philodryas agassizii* is the only species in our study that presents a specialist diet, composed almost exclusively of spiders and scorpions [24]. To verify the similarity of the 3FTX sequences found in the transcriptome of *P. agassizii* with other 3FTX sequences available in databases, we performed a search using BLAST [100], and we obtained more than 70% identity with sequences from *Helicops leopardinus*. This species is a Hidropsini, another Dipsadidae tribe, and it also presents a very specialized diet, which is mainly composed of fish [99]. Likely, 3FTXs are playing relevant roles in the venom of *P. agassizii* and *H. leopardinus*, having a possible correlation with their specialized feeding ecology. Further studies are necessary to confirm this hypothesis.

Regarding some of the less abundant components found in Philodryadini venoms, we have identified CNP, PLB, WAP and Kunitz as noteworthy elements of venom gland transcriptomes. In our analysis, CNP represents 6% of the total transcripts of one individual of *P. olfersii* (SB0001). CNP was already identified in the venom transcriptome of *P. olfersii* in previous studies [54], and it is a potent vasodilator and a hypotensive agent, sharing a common origin with precursors of CNP/BPPs from Viperidae and Elapidae [54]. PLB was found in a small percentage in all transcriptomes and in proteomic profiles. This transcript codes for high molecular weight enzymes, which exhibit hemolytic and cytotoxic activity [101]. Low levels of WAP were present in the transcriptomes of *Philodryas* and *Chlorosoma*. These peptides have approximately 50 amino acids and are generally found in species of Elapidae [102,103]. They have been identified in many snake tissues and were recently found in the transcriptome of the venom gland of *Phalotris mertensi*, a rear-fanged snake that belongs to Elapomorphini, another Dipsadidae tribe [104]. WAP appears to have a protease inhibitory function, including roles in the innate immune system and antibacterial effects [103,105]. WAP can be found associated with two Kunitz domains (inhibitors of serine proteinases) that correspond to the papillin-like protein, which is conserved in reptiles [106]. These minor represented proteins have arguable functions as toxins for snakes in general, being eventually relevant only to some specific taxa.

Overall, our results show a remarkable variability in the venom composition among the three analyzed genera of Philodryadini. Venoms of species of *Philodryas* are predominantly composed of SVMPs of class P-III, whereas species of *Chlorosoma* exhibit a large proportion of seMMPs-9 and species of *Xenoxybelis* present CRISPs as the main component. In this regard, the mechanisms that cause the peculiar expression of CRISPs in dipsadids deserve a better investigation, since the literature still lacks information that could correlate CRISPs as toxic weapons for prey capture.

Conversely, little intraspecific variation was observed within species of Philodryadini. The transcriptome of three individuals of *P. olfersii* showed a very similar general composition, despite their differences in sex, ontogenetic stage and geographic origin. This pattern of high levels of intrageneric divergence, interspecific exclusive profiles and intraspecific homogeneity among the venom composition of Philodryadini may indicate selective forces constraining the individual variability of specific and strongly selected venoms. The specificity of some Philodryadini venoms is better exemplified by the presence of 3FTX in *P. agassizii*, the only species with a specialist diet. Since venom is assumed to play a fundamental role in prey capture in snakes, aspects of feeding ecology may be modulating the evolution of parameters that drive the venom composition in dipsadids, as was previously suggested for vipers [26]. Finally, we emphasize that even if the profiles of the transcripts expressed in the venom gland will not always reflect the same proteins present in the venom, the comparison between both datasets allows for the increase in the reliability of the proteomic result.

## 4. Conclusions

We presented a comprehensive description of the composition and a standardized functional characterization of the venoms produced by snakes of the South American tribe Philodryadini. Our results include complete transcriptomes and proteomes of four species belonging to the genus *Philodryas* (*P. agassizii*, *P. olfersii*, *P. patagoniensis* and *P. nattereri*), two species of the genus *Chlorosoma* (*C. laticeps* and *C. viridissimum*) and the transcriptome from the venom gland of *Xenoxybelis argenteus* and *P. mattogrossensis.* This effort constitutes a substantial addition to previous studies that focused mostly on *P. olfersii* and *P. patagoniensis,* contributing to a more comprehensive analysis of this group of snakes.

In addition to the genetic and morphological differences described in the literature, we observed in our study that the genera *Philodryas*, *Chlorosoma* and *Xenoxybelis* can be also differentiated by the biochemical composition of their venoms. *Chlorosoma* showed a high abundance of seMMPs-9, whereas SVMPs are the most expressed toxin for *Philodryas* and CRISPs are the most abundant component in the venom of *Xenoxybelis*. Our results pave new roads for further studies addressing the evolution of the venom phenotype in Dipsadidae, a group of snakes recognized by their extensive physiological and ecological diversity.

## 5. Materials and Methods

### 5.1. Snakes and Venoms

In total, nineteen snakes belonging to the Philodryadini tribe were collected in different regions of Brazil (ICMBio permits 56576, 57585, and 66597), of which ten specimens were female, eight were males and one we could not identify the sex. Most individuals were adults, except *P. olfersii* (SB0132), *P. patagoniensis* (SB0255) and *P. nattereri* (SB0307), who were juveniles (Table 2).

The individuals were from the South, Southeast, Midwest and North regions of Brazil (Appendix A). The snout–vent length (SVL) variation was 728 mm for *P. olfersii* adult females, and for adult males the SVLs was 565–750 mm. *P. nattereri* females presented SVLs of 468–1190 mm. Males of *P. patagoniensis* showed 525–882 mm SVL, and the female of this same species showed 426 mm SVL. All *X. argenteus* snakes were female and presented 584 and 797 mm SVL. Species with only one individual were *P. agassizii*, *P. mattogrossensis*, *C. laticeps*, *C. viridissimum* and *X. argenteus*, where the values of their SVLs were 265 mm, 930 mm, 884 mm, 808 mm and 584 mm, respectively. After the experiments, the animals were deposited at the Herpetology Collection of Instituto Butantan. Their venom was extracted using pilocarpin on sedated individuals as described previously [103]. Four days after extraction, their venom glands were surgically collected and stored in RNAlater (Invitrogen, Life Technologies Corp., Waltham, MA, USA) at −80 °C. All protocols were certified by the ethical committee of Instituto Butantan (CEUA No. 4479020217).

### 5.2. Electrophoresis in Polyacrylamide Gel in the Presence of SDS (SDS-PAGE)

The samples of *Philodryas* venom pools and *Chlorosoma* individuals (15 μg) were analyzed by one-dimensional electrophoresis in 12.5% polyacrylamide gel according to the method described by Laemmli (1970) [107]. The individuals *Philodryas mattogrossensis*, *Philodryas agassizii* and *Xenoxybelis argenteus* were not subjected to analysis due to insufficient amounts of venom. The samples were submitted to non-reducing conditions (Tris-HCl 0.125 M, pH 6.8, plus glycerol at 10%, 2% SDS and 0.001% bromophenol blue) and reduced conditions (solution equivalent to that described above, with the addition of 2-mercaptoethanol 58.3 mM). Then, the samples were boiled for 5 min, and after application in the gel they were submitted to electrophoresis. The amperage for electrophoresis was 35 mA to 180 V. After the end of the run, the samples were stained with Coomassie blue R-250 solution at 0.2% in 50% methanol. The bleaching solution used consists of 30% ethanol and 10% acetic acid. The proteins used as molecular mass standard (GE Healthcare, 2006) were phosphorylase B (97 kDa), albumin (66 kDa), ovalbumin (45 kDa), carbonic anhydrase (30 kDa), trypsin inhibitor (20.1 kDa) and α-lactalbumin (14.4 kDa).

### 5.3. Transcriptomic Analysis

Ten individuals venom glands were pulverized in a Precellys 24 homogenizer and RNA was extracted with TRIZOL (Thermo Fisher Scientific, São Paulo, Brazil) following the method described by Chomczynski and Sacchi (1987) [108], with minor modifications. Total RNA was quantified by the Quant-iTTMRiboGreen RNA reagent and kit (Thermo Fisher Scientific, São Paulo, Brazil). Quality control of the extracted RNA was performed in an Agilent 2100 bioanalyzer using the Agilent RNA 6000 Nano Kit to verify the integrity of total RNA through band discrimination corresponding to fractions 18S and 28S of total RNA. cDNA libraries were prepared for each individual sample. About 1 μg of total RNA was used with the Illumina TruSeq Stranded RNA HT Kit (Illumina). Fragment size distribution was evaluated by microfluidic gel electrophoresis in the bioanalyzer device (Agilent 2100) using the Agilent DNA 1000 Kit according to the manufacturer’s protocol. Quantification of each library was then performed by real-time PCR using the KAPA SYBR FAST Universal qPCR Kit, according to the manufacturer’s protocol, using the StepOnePlusTM Real-Time PCR System. All procedures with RNA were performed with RNAse-free tubes and filter tips and using water treated with diethylpyrocarbonate (DEPC, Sigma). Aliquots of each cDNA library were diluted to a concentration of 2 nM. Then, a pool of all samples with 5 μL from each library was prepared, and the pool concentration was determined by real-time PCR. cDNA libraries were sequenced in the Illumina HiSeq 1500 System (Hayward, CA, USA) using a rapid paired-end *flowcell* in 2 × 150 bp configuration.

#### 5.3.1. Transcriptome Assembly

Initially, the reads generated were verified for cross-contamination between samples sequenced in the same *flowcell* using an internal Python script, as previously described by Hofmann et al. (2018) [109]. Then, the adapter sequences and low-quality reads were removed using TrimGalore [110]. The reads were combined using PEAR software to produce larger sequences through a merging process [111]. Merged reads were grouped, resulting in more than 11 million reads for each sample, with the exception of individuals SB0001, SB0026, SB1590 and SB1881, with more than 2 million merged reads for each of these individuals. Several assemblers were used to maximize the chances of obtaining complete toxin transcripts: Trinity v2.4.0 [112], SeqMan NGen v. 14 (DNAStar, Inc., Madison, WI, EUA), SPAdes [113], Bridger [114] and an internal assembler denominated “Extender” [115]. Trinity assemblies were performed with a minimum contig length of 200 bp and a k-mer size of 31, for the SeqMan NGen with default settings (k-mer size of 21), SPAdes with k-mer values of 31, 75 and 127 and Bridger with a value of 30 k-mer, while Extender was applied by setting two minimum overlaps for extension (120 bp and 150 bp) and the minimum quality score of at least 20 in all base positions for a read to be considered for extension. Overall, we had an average of 176,000 contigs per individual. Raw transcriptomic data are available at NCBI’s GenBank under Bioproject accession number PRJNA910168. Curated sequences (CDS and translated proteins) generated in this work are available in Appendix A.

#### 5.3.2. Toxins Annotation

The contigs of each individual were aligned against the cured databases of toxin sequences obtained in our previous and GenBank studies using BLASTn [100]. The manual curation of complete ORFs was performed on Geneious software (v.2020.0.5) in conjunction with SignalP Server 6.0 (DTU Health Tech) for checking the presence of the signal peptide region and HMMER (EMBL-EBI) for analyzing sequence domains. In addition, coverage analysis was done by manually examining the read mapping profile and using ChimeraCheck software [109], followed by removal of possible chimeric sequences. Nonredundant sequences that passed all filters were incorporated into a final master set for Philodryadini. The expression of each contig was estimated using the RNA-seq by Expectation-Maximization (RSEM) software [116] after mapping reads from each sample using Bowtie2, and estimates were made in transcripts per million (TPM) [117].

### 5.4. Analyses of Venoms by Mass Spectrometry (LC-MS/MS)

Shotgun liquid chromatography–tandem mass spectrometry analyses of the individual venoms or pools from 7 species were performed by the University of São Paulo BIOMASS—Core Facility for Scientific Research-USP (CEFAP-USP), following the methods by Freitas-de-Sousa and collaborators (2020) [35]. The mass spectra were processed and searched in an internal database using the Mascot research tools (Matrix Science, London, UK; version 2.4.1). The database used to identify the MS/MS spectra is composed of putative sequences from the combination of transcriptomes from all Philodryadini species analyzed here. We have validated peptide and protein identifications using Scaffold (v.5.2.1; Proteome Software Inc.) and we used the same criteria described by Freitas-de-Sousa et al. (2020) [35] for peptide/protein identification and validation. The quantification of proteins related to the general abundance in each venom was based on normalized total spectra counts, and for identifications at the isoform level, venom analyses were expressed as normalized exclusive unique spectral counts, corresponding to the number of spectra attributed to proteotypic peptides of a given protein entry present in the database.

### 5.5. Functional Analysis of Venoms

#### 5.5.1. Enzymatic Assays on Synthetic Substrates

Enzymatic activities on synthetic substrates of metalloproteinases (SVMPs) and serine proteinases (SVSPs) were evaluated according to the methods described by Knittel et al. (2016) [118]. For the assay with SVMPs, 10 μg of venom diluted in reaction buffer (50 mM Tris-HCl, 10 mM CaCl_2_, 150 mM NaCl and 0.05% Brij 35-Sigma-Aldrich, pH 7.5) were incubated with 50 μM of FRET (fluorescence resonance energy transfer) substrate Abz-AGLA-EDDnp (Peptide International), and enzymatic reactions were monitored in a SpectraMax^®^ M2 fluorimeter (Molecular Devices, San Jose, CA, USA) with excitation at 340 nm and emission at 415 nm at 37 °C in kinetic mode. As a positive control, *Bothrops jararaca* venom (10 μg) was used. Results were expressed in relative fluorescence units (RFU/min/μg). The SVSPs activity of venom samples was determined using 800 μM Nα-benzoyl-arginyl-p-nitroanide substrate (L-BAPNA–Sigma Life Science) incubated with 30 μg of venom diluted in 10 mM Tris-HCl buffer, pH 7.5 for 40 min at 37 °C. As a positive control, we used *Bothrops neuwiedi* venom (30 μg). Activity was determined according to absorbance at 425 nm and expressed as absorbance/min/μg of venom.

#### 5.5.2. Gelatinolytic Assay on SDS-PAGE Gel

The gelatinolytic assay of venoms was performed using 0.5 μg of each venom diluted in sample buffer (0.25 M Tris-HCl, pH 6.8, 10% glycerol, 2% SDS and 0.001% bromophenol blue) in the absence of reducing agent. The samples were applied on a 12.5% SDS-PAGE gel with 2 mg/mL of gelatin, and electrophoretic separation was performed at 160 V and 35 mA. After this procedure, the SDS was removed by 3 washes of 20 min at room temperature in a 2.5% Triton X-100 solution. Gels were incubated at 37 °C for 18 h in 50 mM Tris-HCl, pH 7.4, containing 5 mM CaCl_2_. For the inhibitor test, 20 mM EDTA was added to the incubation buffer. After this period, we followed the same staining and bleaching protocol described in Section 5.2.

#### 5.5.3. Gelatinolytic Assay against 1,10-Phenanthroline and PMSF Inhibitors

In order to further explore the gelatinolytic capacity of the venoms, another assay was performed, this time in a 96-well plate using 10 μg of each venom diluted in the same reaction buffer used in the SVMPs assay. As a positive control, we used 30 μg of *Bothrops jararaca* venom. The inhibitors used in the assay were 1,10-phenanthroline and phenylmethylsulfonyl fluoride (PMSF) diluted in DMSO 1M at a concentration of 100 mM per well. Venoms and venom solutions with inhibitors were pre-incubated at 37 °C for 15 min. After incubation, they were centrifuged at 10.000 rpm for 10 min and the assay was performed with the addition of 1 mg/mL of porcine skin gelatin conjugated with fluorescein (Invitrogen, Waltham, MA, USA). The assay activity was measured in a spectrofluorimeter with an excitation wavelength of 495 nm and fluorescence emission of 515 nm and determined by relative fluorescence units (RFU/min/μg).

### 5.6. Identification of Gel Interest Bands by Mass Spectrometry (Q-TOF-LC-MS)

The in-gel digestion was conducted according to Westermeier, Naven and Höpker (2008) [119] with small modifications. Firstly, the bands were submitted to a destain solution of 50 mM ammonium bicarbonate in 40% acetonitrile (ACN). The bands were washed under centrifugation at 1500 rpm, 4 °C for 30 min until complete removal of Coomassie Brilliant Blue and the supernatant was removed. ACN was added to the samples until the bands became completely white in color, and samples were dried at 30 °C in a speed vac system. For the reduction step, 10 µL of 10 mM dithiothreitol (in 25 mM ammonium bicarbonate) was added to the bands, and all samples were subsequently incubated at 60 °C for 30 min. For alkylation of proteins, 10 µL of 100 mM iodoacetamide (in 25 mM ammonium bicarbonate) was added to the bands, and all samples were incubated at room temperature for 30 min in the absence of light. Samples had their supernatant removed and were dehydrated by adding acetonitrile. Thereafter, 20 µL of trypsin solution (50 ng/µL in 50 mM ammonium bicarbonate) was added to each sample, and gel pieces were covered with 50 mM ammonium bicarbonate and incubated overnight at 30 °C. Samples were digested for 16 h under 30 °C incubation. For quenching the reaction, 5 µL of 10% formic acid solution was added to the samples. The supernatant was removed and dispensed in a separate tube. Finally, each sample was suspended in 75 µL of 40% ACN/0.1% formic acid and submitted to centrifugation at 1500 rpm, 25 °C /30 min. We repeated this step three times and combined the supernatants of the same samples. Lastly, samples were concentrated in a vacuum system and suspended in 20 µL of 2% ACN/0.1% formic acid. The tryptic peptides were analyzed by liquid chromatography-mass spectrometry (LC-MS/MS) using an LCMS-9030 Q-TOF (Shimadzu, Japan). Briefly, samples were dried, resuspended in 0.1% formic acid and loaded onto a trap C18 column (nanoEase M/Z Peptide BEH C18 130Å, 300 µm × 50 mm, 5 µm) and analytical C18 column (ACQUITY UPLC M-Class CSH C18, 130Å, 300 µm × 100 mm, 1.7 µm) operating with a solvent system of water and 0.1% formic acid (*v*/*v*) (solvent A) or ACN and 0.1% formic acid (*v*/*v*) (solvent B). The column was eluted at a constant flow rate of 10 µL/min with a linear gradient of solvent B (3% to 45% /50 min). The MS spectra were acquired under the positive mode and collected in a range of m/z 100–2000, scan time 0.5 s, ramp collision energy: 25 to 50 V. The MS/MS spectra were collected in the range of 50–2000 m/z, scan 0.25 s, top 5 MS/MS. The processing of these proteomic data was performed from the raw data generated by LCD LCMSolution Shimadzu, which were converted into mzXml by the LCMSolution tool (PRIDE) and then loaded into Peaks Studio V7.0 (BSI, Canada, Waterloo). Data were processed according to the following parameters: MS and MS/MS error mass were 0.07 Da; methionine oxidation and carbamidomethylation used as variable and fixed modification, respectively; trypsin as cleaving enzyme; maximum missed cleavages (3), maximum variable PTMs per peptide (3) and non-specific cleavage (both); the false discovery rate was adjusted to ≤ 0.5%; only proteins with score ≥20 and containing at least 1 unique peptide were considered in this study. Data were analyzed against a final master set generated for the Philodryadini tribe, although broad searches against the whole UniProt were performed, as well as quality controls (Appendix A).

### 5.7. Statistical Analyses

Results were expressed as mean ± standard deviation. Statistical analysis was performed using GraphPad Prism software (GraphPad Software, Inc., San Diego, CA, USA). Data normality was assessed using the Kolmogorov–Smirnov normality test. Differences between two groups were analyzed using the t-Student test (unpaired or non-parametric test). Differences between groups were considered statistically significant at * *p* < 0.05; ** *p* < 0.01; *** *p* < 0.001; **** *p* < 0.0001.

## Figures and Tables

**Figure 1 toxins-15-00415-f001:**
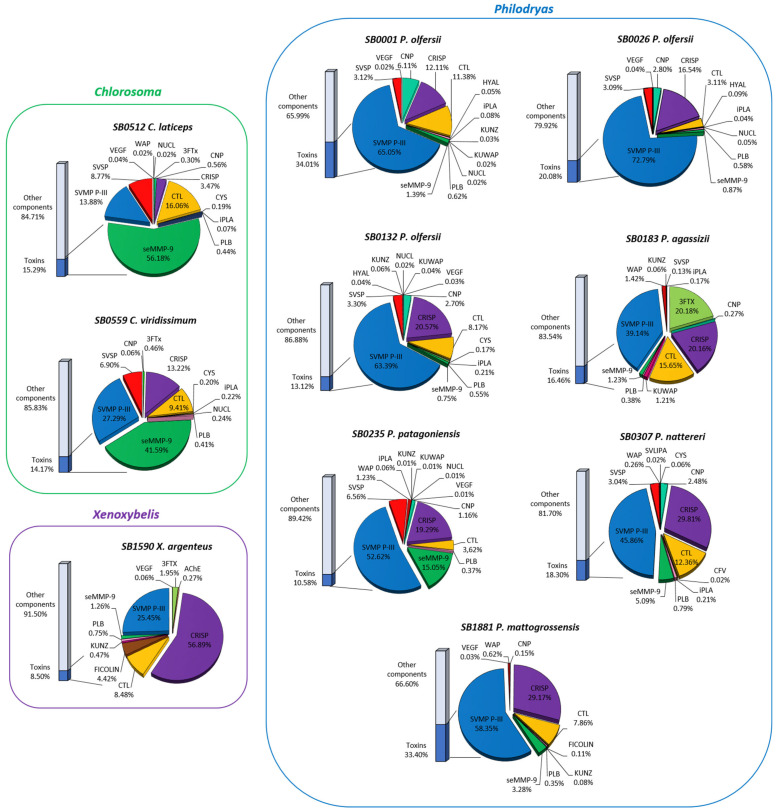
Proportion of toxin families expressed in the venom gland transcriptomes of seven specimens of *Philodryas*, two specimens of *Chlorosoma* and one of *Xenoxybelis argenteus*. Percentages represent the sum of TPM (transcripts per million reads) for each category. 3FTX (three-finger toxin), AChE (acetylcholinesterase), CNP (C-type natriuretic peptide), CRISP (cysteine-rich secretory protein), CTL (C-type lectins), CVF (cobra venom factor), CYS (cystatin), FICOLIN (ficolin), KUNZ (BPTI/KUNITZ inhibitor domain), KUWAP (Ku-wap-fusin), iPLA (phospholipase inhibitor), PLB (phospholipase B), seMMP-9 (snake endogenous matrix metalloproteinase 9), SVMP P-III (snake venom metalloproteinase P-III class), SVSP (snake venom serine proteinase), NUCL (5-nucleotidase), VEGF (vascular endothelial growth factor) and WAP (waprin whey acidic protein-type). SB codes correspond to the identification of individuals.

**Figure 2 toxins-15-00415-f002:**
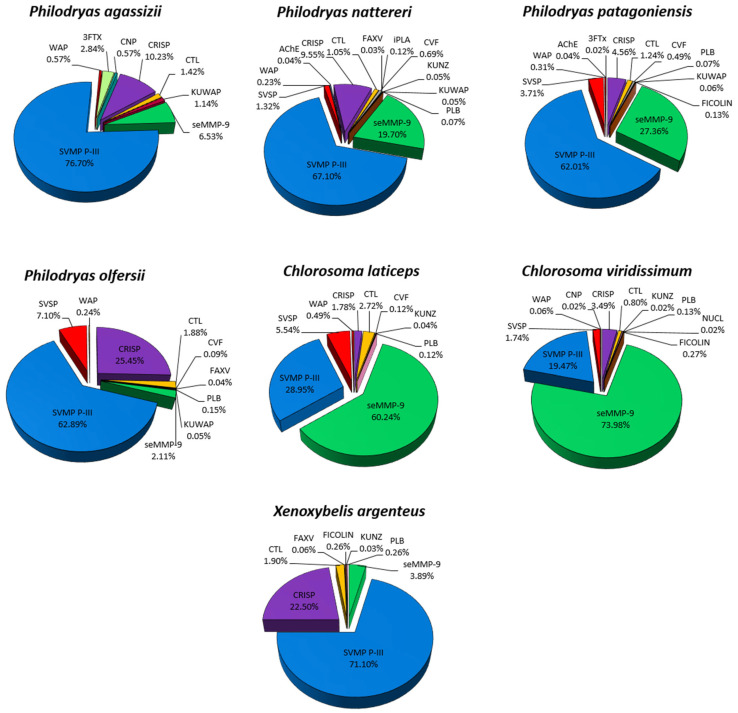
Proportion of toxin families in the venom pools of Philodryadini snakes identified from proteomic analysis by mass spectrometry (LC-MS/MS). The normalized total spectra count (TSC) defined the relative expression of each toxin family. 3FTX (three-finger toxin), CNP (C-type natriuretic peptide), CRISP (cysteine-rich secretory protein), CTL (C-type lectins), FICOLIN (ficolin), KUNZ (BPTI/KUNITZ inhibitor domain), KUWAP (Ku-wap-fusin), iPLA (phospholipase inhibitor), PLB (phospholipase B), seMMP-9 (snake endogenous matrix metalloproteinase 9), SVMP P-III (snake venom metalloproteinase P-III class), SVSP (snake venom serine proteinase) and WAP (waprin whey acidic protein-type).

**Figure 3 toxins-15-00415-f003:**
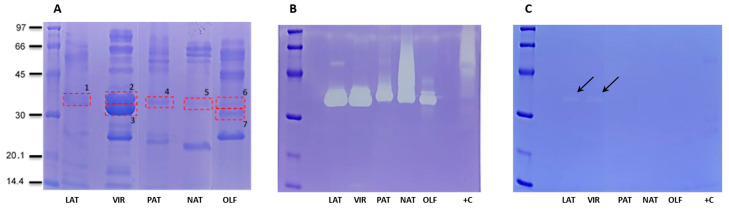
Gelatinolytic activity of Philodryadini venoms through gel zymography. (**A**). Samples (0.5 μg of venom) were run in non-reducing conditions on SDS-PAGE 12.5%. The gel was stained with bright blue Coomassie R-250. The dashed red squares correspond to the bands excised from the gels for LC-MS/MS analysis (Appendix A). (**B**). Samples were run as described above but in SDS-PAGE 12.5% containing 20 mg/mL of gelatin. After electrophoresis, the gel was washed with Triton X-100 (2.5%) and incubated overnight in reaction buffer (20 mM Tris, 0.5 mM CaCl_2_, pH 7.4) at 37 °C. The gel was then rinsed with distilled water and stained with bright blue Coomassie R-250. The proteolytic activity produced white/transparent bands on a blue background due to the gelatin hydrolysis. (**C**). The same procedure as in B was used but 20 mM EDTA was included in the reaction buffer to inhibit the metalloproteinase activity on the gelatin substrate. Black arrows indicate the faint hydrolysis bands. *Chlorosoma laticeps* (LAT), *Chlorosoma viridissimum* (VIR), *Philodryas patagoniensis* (PAT), *Philodryas nattereri* (NAT), *Philodryas olfersii* (OLF) and positive control *Bothrops jararaca* (+C).

**Figure 4 toxins-15-00415-f004:**
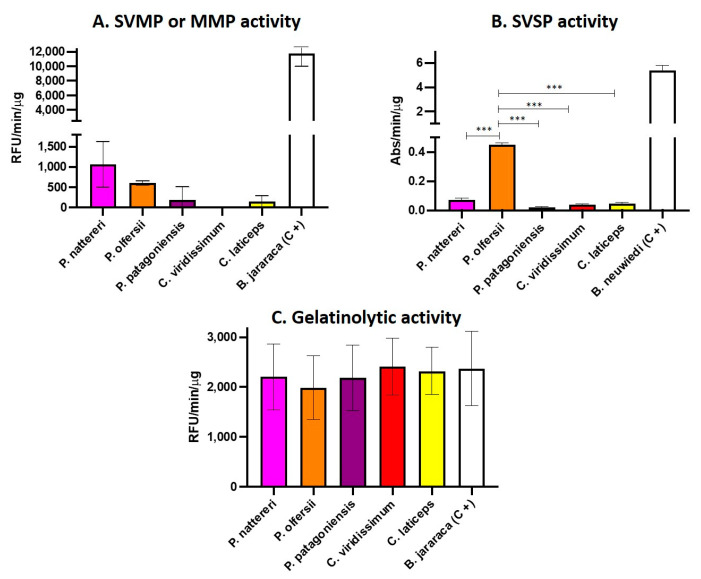
Functional activity of *Philodryas* and *Chlorosoma* snake venoms according to enzymatic assays on synthetic substrates for SVMPs and SVSPs. (**A**). Metalloproteinase (SVMP or MMP) activity was measured by fluorimetry using Abz-AGLA-EDDnp substrate and 10 μg of venom. The results are presented as the ratio of relative fluorescence units (RFU), by time (min) and by the amount of protein used in the assay (μg). (**B**). Serine proteinase (SVSP) activity was assessed by hydrolysis of benzoyl-arginine-p-nitroanide (L-BAPNA) using 30 μg of venom. The results are presented by the absorbance ratio (Abs), by the time (min) and by the amount of protein used in the assay (μg). (**C**). The gelatinolytic activity was measured through fluorometric assays using as substrate gelatin from synthetic pig skin conjugated with fluorescein, incubated with 10 μg of Philodryadini venoms or 30 μg of *B. jararaca* venom as a positive control. The results are presented by the ratio of relative fluorescence units (RFU), by time (min) and by the amount of protein used in the assay (μg). Value *** *p* < 0.001 corresponds to statistically significant.

**Figure 5 toxins-15-00415-f005:**
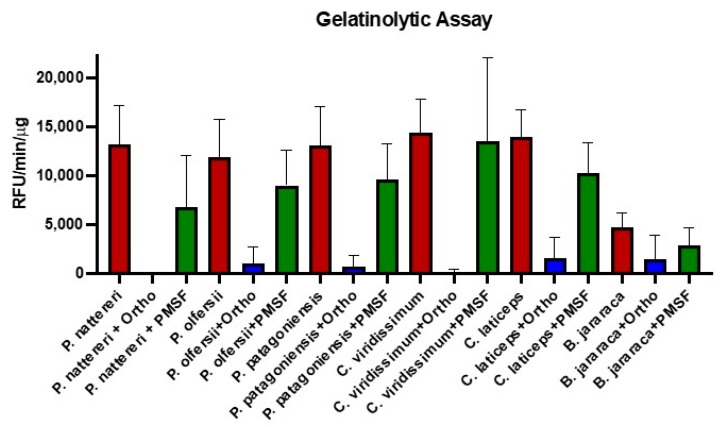
Effects of metalloproteinase and serine proteinase inhibitors in the gelatinolytic activity. The gelatinolytic activity was measured through fluorometric assays using as substrate gelatin from synthetic pig skin conjugated with fluorescein, incubated with 10 μg of Philodryadini venoms or 30 μg of *Bothrops jararaca* venom pre-incubated with 100mM of 1,10-phenanthroline and PMSF inhibitors. Species-only names correspond to applied venom samples diluted in reaction buffer. Ortho: 1,10-phenanthroline. PMSF: phenylmethylsulfonyl fluoride.

**Table 1 toxins-15-00415-t001:** Peptides from Philodryadini venoms identified with the highest score in each band of interest.

Philodryadini Database					
Species	Band	−10lgP	Coverage	Peptides	Identification
*C. laticeps*	1	404.70	49	56	seMMP9-SB0512_CLATSeMMP007
*C. viridissimum*	2	438.68	45	70	seMMP9-SB0559_CVIRSeMMP011
*C. viridissimum*	3	496.05	55	94	seMMP9-SB0559_CVIRSeMMP018
*P. patagoniensis*	4	442.28	51	57	seMMP9-SB0235_PPATSeMMP001
*P. nattereri*	5	302.95	30	22	seMMP9-SB0307_PNATSeMMP001
*P. olfersii*	6	289.52	28	18	seMMP9-SB0559_CVIRSeMMP011
*P. olfersii*	7	293.78	43	13	SVSP-SB0001_POLFSVSP002

−10lgp: it means that the score is −10 times the common logarithm of the *p*-value.

**Table 2 toxins-15-00415-t002:** Information on all Philodryadini snakes used in this study collected in Brazil between 2016 and 2021 in different seasons.

Species	Number Biota Project	Sex	Reproductive Status	Sample Origin	SVL (mm)	Use
*Philodryas olfersii*	SB0001	F	adult	Pirenópolis/GO	728	T
	SB0026	M	adult	Porto Alegre/RS	565	P
	SB0130	M	adult	Rio Grande do Sul	774	P
	SB0131	F	adult	Rio Grande do Sul	782	P
	SB0132	Nd	juvenile	Aquidauana/MS	352	T/P
	SB0370	M	adult	Alegrete/RS	750	P
*Philodryas patagoniensis*	SB0136	M	adult	Rio Grande do Sul	882	P
	SB0235	M	adult	Rosário do Sul/RS	525	T/P
	SB0255	F	juvenile	Bom Jardim da Serra/SC	426	P
*Philodryas nattereri*	SB0307	F	juvenile	Nd	468	T
	SB0707	F	adult	Jaíba/MG	1095	P
	SB1027	F	adult	Araguari/MG	1155	P
	SB1028	F	adult	Araguari/MG	1190	P
*Philodryas agassizii*	SB0183	M	adult	Painel/SC	265	T/P
*Philodryas mattogrossensis*	SB1881	F	adult	Nova Ponte/MG	930	T
*Chlorosoma lacticeps*	SB0512	M	adult	Sooretama/ES	844	T/P
*Chlorosoma viridissimum*	SB0559	M	adult	Sena Madureira/AC	808	T/P
*Xenoxybelis argenteus*	SB1590	F	adult	Mâncio Lima/AC	584	T
	SB1934	F	adult	Cruzeiro do Sul/AC	797	P

SVL—snout–vent length; P—proteomics; T—transcriptomics; Nd—not determined.

## Data Availability

Raw transcriptomic data are available at NCBI’s GenBank under Bioproject accession number PRJNA910168 at https://www.ncbi.nlm.nih.gov/bioproject/?term=PRJNA910168 (accessed on 8 December 2022).

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
