# Peer review of "The Venom Composition of the Snake Tribe Philodryadini: ‘Omic’ Techniques Reveal Intergeneric Variability among South American Racers"

_toxins, 2023, doi:10.3390/toxins15070415_

Round 1
Reviewer 1 Report
The venom composition of the snake tribe Philodryadini: 2 ‘Omic’ techniques reveal intergeneric variability among South 3 American racers
Summary: This article provides an overview of venom composition and venom gland gene expression from Philodryadini.
Results
The results sections could be improved by providing actual data and the differences in the data. For example the authors use vague terminology such as “low” or ‘increased” rather than providing the numerical values and standard deviations. See line 196 260, 268
The standard deviation of the percentages should be provided unless the RNA and venom was pooled, which is not clear in the manuscript.
Methods
5.3
Line 574: “RNA was extracted…..” This is too vague and does not specifically explain where the RNA was extracted. This is inferred in table 2 but could be more specifically addressed in this section.
This overall section is too vague and should include number of individual venom glands or pooled samples, DNAse treatment, evaluation of RNA integrity.
5.3.1
Line 588-593. Expanding this section in detail would make the manuscript more clear to the reader. Number of contigs? Raw reads? What are the default settings of Trinity? Total number of assembled bases? Although some is supplied in supplementary information, adding it to the manuscript text would improve the read.
5.3.2
What is RSEM? Some readers may not be familiar with this acronym. Cut off e-values?
Author Response
We appreciate the reviewer's suggestions and respond to them below.
Results
The results sections could be improved by providing actual data and the differences in the
data. For example the authors use vague terminology such as “low” or ‘increased” rather than
providing the numerical values and standard deviations. See line 196 260, 268
The standard deviation of the percentages should be provided unless the RNA and venom was
pooled, which is not clear in the manuscript.
Answer: In our study pools of venoms were used and for most species only one venom gland, so it is not possible to present the standard deviation of the percentages. It was included in the Results text that venom pools were used and is also described in item 5.4 and available in Table 2.
Methods
5.3
Line 574: “RNA was extracted…..” This is too vague and does not specifically explain where
the RNA was extracted. This is inferred in table 2 but could be more specifically addressed in
this section.
This overall section is too vague and should include number of individual venom glands or
pooled samples, DNAse treatment, evaluation of RNA integrity.
Answer: We have included them in the text.
5.3.1
Line 588-593. Expanding this section in detail would make the manuscript more clear to the
reader. Number of contigs? Raw reads? What are the default settings of Trinity? Total number
of assembled bases? Although some is supplied in supplementary information, adding it to the
manuscript text would improve the read.
Answer: As requested, have included them in the text.
5.3.2
What is RSEM? Some readers may not be familiar with this acronym. Cut off e-values?
Answer: We have included them in the text.
Reviewer 2 Report
The authors have characterised the venoms of 8 species of South American snakes of the family Dipsadidae, tribe Philodryadini, genera Philodryas, Chlorosoma, and Xenoxybelis. The venoms were compared both transcriptomically and proteomically. The venoms of Phylodryas and Xenoxybelis were shown to be SVMP dominant, while Chlorosoma was MMP dominant. All genera also showed high expression of CRiSP. The study makes an important contribution to our knowledge of the understudied rear-fanged colubrids and the data should be published and made available to other snake venom researchers. Some minor editing is required and there were a couple of important citation omissions which need to added. These changes are in the attached word document. Provided these changes are made I support publication of the manuscript.

English adequate, have attached some minor editing that needs to be addressed.
Author Response
We appreciate the reviewer's suggestions and respond to them below.
Line 45 –“seldom” instead of “seldomly”
Answer: We have adjusted in the text.
Line 49- “present” instead of “presents”
Answer: We have adjusted in the text.
Line 63 – “occasionally” instead of “eventually”
Answer: We have adjusted in the text.
Line 67 – there are far more recent and comprehensive reviews of snake venom than this
“A review and database of snake venom proteomes” 2017. Tasoulis and Isbister
“A current perspective on snake venom and constituent protein families” 2023. Tasoulis and Isbister
Answer: We have included them in the text.
Line 67- change to “possible digestion of prey” it has not been proven to be the case
Answer: We have included them in the text.
Line 209- change “called our attention” to “attracted our attention
Answer: We have adjusted in the text.
Reviewer 3 Report
Dear Authors,
Please see attached suggestions and comments.
Best regards,

Author Response
We appreciate the reviewer's suggestions and respond to them below.
1-The authors must address the significant discrepancies observed in the transcriptome
and corresponding proteome of SVMP III in the venoms of P. agassizii (39.1% : 76.7%)
and seMMP-9 in C. viridissimum (41.6% : 74%) as depicted in Figure 1 and 2.
Answer: Thank you for the suggestion. A possible explanation for this discrepancy was added in the text. Some mechanisms that affect protein translation could interfere with the amount of venom observed in the proteome. However, in our study we analyzed for some species a venom pool but only one venom gland, so a more robust analysis is needed to confirm this hypothesis.
2. The authors must improve the label resolution in Figure 3 for better clarity and
readability.
Answer: The labels in figure 03 were improved.
Round 2
Reviewer 3 Report
Dear Author(s),
Thank you for the revision.
Sincerely,